# Interpreting Environmental Impacts Resulting from Fruit Cultivation in a Business Innovation Perspective

**Marco Medici \***[ID]**, Maurizio Canavari**[ID] **and Moreno Toselli**[ID]

Department of Agricultural and Food Sciences, Alma Mater Studiorum-University of Bologna,
40127 Bologna, Italy; maurizio.canavari@unibo.it (M.C.); moreno.toselli@unibo.it (M.T.)
**\*** Correspondence: m.medici@unibo.it

**Abstract:** Sustainability of food production is a major concern today. This study assessed the environmental impact of fruit production and discussed business implications for sustainability. Data were collected from three agricultural enterprises growing six species of fruit, extending over a total of 34 hectares, and producing roughly one thousand tons of fruit per year. The results of the life-cycle assessment (LCA) showed that several production activities heavily impact the environment: in descending order of absolute terms, fruit refrigeration, agronomic operations, irrigation, and fertilizer use were recognized as the most impacting. Other activities, including agrochemical applications, planting, and plastic use for harvesting and packaging, showed overall lower impacts. The high environmental impact associated with most of the production activities emphasizes the need to make the primary food production cleaner, more resource-efficient, and less energy-intensive. Affordable incremental innovations able to reshape the way business is conducted in the context of primary food production are proposed, mainly relying on process rationalization and digital switchover. The analysis of the business path toward increased sustainability involves strategic issues, ranging from the reshaping of production processes to relationships with consumers, affecting value proposition, creation, and capture.

**Keywords:** business models; innovation; resource efficiency; sustainability; environmental assessment; fruit production; cleaner production; life cycle assessment (LCA); carbon footprint

## 1. Introduction

The agriculture and food sector is of primary importance to human health and well-being. On the other hand, it generates several adverse environmental effects that harm natural resources, constituting the input of food production. These effects are the by-products of food production, which are largely subjected to inefficiencies in processes, energy, and input use. The importance of integrating sustainability into the production process has caught the attention of practitioners and researchers, in several fields, over the last few decades [1]. The need for sustainable practices has been recognized in the face of challenges, such as achieving cost-effective production and reducing energy consumption [2].

This kind of issue was recently addressed in the context of a sustainable business model innovation, in which scholars, practitioners, and, generally, food system stakeholders, started to pay attention in the last decade [3]. This concept concerns the way business is performed by generating competitive advantages while contributing positively to enterprises, the environment, and society [4]. Within such a framework, it is widely recognized that positive business changes can be stimulated by adopting various tools for sustainable business thinking. These tools include life-cycle assessment (LCA), as defined by the International Organization for Standardization (ISO) [5,6], eco-design [7], eco-ideation [8], the business model canvas, as readapted by Bocken et al. [9], and stakeholder analysis [10]. The adoption

of these practices, especially in large manufacturing firms, has so far encompassed the elimination of pollution from processes, the minimization of waste material, waste treatment, the reduction of energy consumption, as well as the attraction of customers, and cost savings [11].

One of the peculiarities distinguishing primary food production from industrial processing is the limited, but significant, scope related to sustainability innovations, as value proposition is essentially based on natural products, with limited to no presence of industrial manufacturing. For this reason, tools, such as LCA, designed to quantify the environmental burden associated to material and energy consumption of a product across its entire life, and can provide exhaustive information to assess current business environmental footprints, as well as production inefficiencies [5]. Besides giving an overview of the existing production processes, LCA can be used to draw out possible solutions to mitigate production impacts, offering a holistic perspective to conduct business model analysis for innovation and pave the way for environmental and economic improvements. This use is especially relevant for small and medium enterprises (SMEs), which generally lag behind large companies in the implementation of sustainability management tools [12]. Facing this problem, we formulated the following research question: which solutions may be implemented at the strategic level to mitigate the environmental impact of production activities and achieve a higher level of sustainability?

In this work, we applied an LCA to fruit production to give a measure of the environmental impact characterizing fruit production activities, while identifying inefficiencies, and finding room for possible business model innovations. We selected a case study in the Emilia-Romagna region (northeast Italy) represented by three large agricultural enterprises growing six species of fruit, extending over 34 hectares, and producing a total of roughly one thousand tons of fruit per year. The study is relevant because, economically, fruit growing is one of the most important agricultural activities in Emilia-Romagna, a region that is considered Italy's main orchard. Fruit growing is also an intensive crop in the area, and it is socially important because of its impact on job opportunities and cultural elements. Its reach goes beyond the basic agricultural activity: once harvested, fruit produce is directly available for consumption or can be further processed to obtain products, such as juices and fruit preserves, an industry that is deeply rooted in the Emilia-Romagna region. The paper is organized as follows. In Section 2, we describe the case study and apply the LCA methodology, reporting the goal and scope definition, and the inventory analysis; LCA results are reported in Section 3. In Section 4, we propose possible technological and process improvements, also discussing business strategy implications. In Section 5, concluding remarks and indications for future research are outlined.

## 2. Materials and Methods

### 2.1. Description of the Case Study

Data were collected in 2019 from three agricultural enterprises in which the following fruit species are cultivated: apricot, nectarine, pear, plum, apple, and kiwi. The orchards are located between Forlì and Faenza, in the Emilia-Romagna region (northeast Italy). Telephone interviews and on-site meetings with farm technicians during fruit cultivation were administered to collect qualitative information and quantitative data. Table 1 summarizes the main characteristics of the orchards. The LCA was performed to assess the environmental impact associated with the fruit production originating from these orchards, measured according to the dedicated technical standard ISO 14040:2006 (principles and framework) [5] and 14044:2006 (requirements and guidelines) [6], following the specified phases: goal and scope definition, inventory analysis (life cycle inventory (LCI) phase), impact assessment (life cycle impact assessment (LCIA) phase), and result interpretation.

### 2.2. Goal and Scope Definition

LCA was performed to measure the environmental impact characterizing the fruit produced in the three agricultural enterprises, with the aim of identifying process inefficiencies and finding room for possible ad-hoc innovations able to make production more efficient, less energy-intensive,

and less dependent from non-renewable energy sources. In addition to this, LCA results will be used for learning and educational purposes. The identification of inefficiencies throughout the production process can make enterprise decision-makers aware of existing environmental burdens, with the LCA itself representing a step towards the comprehension of sustainability management tools by technicians. LCA results can also serve as decision support for individuals, both as citizens or consumers, potentially directly entering in eco-labels or other consumer information from producers (e.g., printed on the packaging) or indirectly through reporting research findings [13]. In this respect, LCA can effectively support consumer decisions in choosing products with a lower environmental impact.

**Table 1.** Main characteristics of the orchards.

|  | **Apricot** | **Nectarine** | **Plum** | **Pear** | **Apple** | **Kiwi** |
|---|---|---|---|---|---|---|
| Variety | Faralia | Romagna Red | September Yummy | Abate Fetel | Rosy Glow | Hayward |
| Training system | Vase | Slender spindle | Slender spindle | Slender spindle | Solax | Pergola |
| Growing surface (ha) | 0.35 | 1.20 | 7.70 | 1.16 | 1.04 | 22.95 |
| Layout (m) | 4.0 × 1.5 | 3.5 × 0.6 | 4.2 × 1.5 | 4.5 × 2.0 | 3.5 × 2.0 | 5.0 × 2.0 |
| Plant density (trees/ha) | 1666 | 4762 | 1587 | 1111 | 1428 | 1000 |
| Year of establishment | 2011 | 2012 | 2013 | 2010 | 2010 | 2017 |
| Orchard life (years) | 15 | 15 | 15 | 15 | 15 | 20 |
| Yearly yield (t/ha) | 21.6 | 27.5 | 59.9 | 22.9 | 58.6 | 16.1 |

The approach adopted for the estimation was cradle-to-farm gate, evaluating the impact of the fruit produced, i.e., immediately available for direct sale or distribution. All of the input used for production, particularly mass and energy flows, have been tracked, considering the following four fruit life cycle phases: fruit cultivation, use of agricultural and energy inputs in agronomic activities, fruit harvest, and fruit storage in cold rooms. Data were collected during 2019 until last harvests were completed, concerning the season 2018–2019.

*2.3. Functional Unit and System Boundaries*

The functional unit (FU) relates to all inputs and outputs in the LCI and consequently, the LCIA profile, defining what is being studied [6]. In this study, 1 kg of each fruit, packaged in a 14 g polyethylene package, was chosen as FU, considering the yields in the season 2018–2019, as such, unit lends itself to possible downstream consumer behavior analysis. Therefore, data provided by enterprise technicians, expressed as unit per hectare of the orchard, were traced back to a yield expression (kg per hectare) to make it consistent with FU. The system boundaries were defined, as shown in Figure 1. In addition to the processes outlined in Figure 1, planting of fruit trees was included within the system boundaries and weighted, considering a 15-year (20 years for kiwi) orchard lifetime. The orchard end-of-life was modeled in the same way, considering the use of machinery for tree pruning. The mass of tree by-products was assumed equal to 50 kg for apricot and plum, 70 kg for nectarine, 20 kg for pear, 40 kg for kiwi, and 13 for apple, which is consistent with the tree mass assessment reported in [14].

*2.4. Inventory Analysis*

The inventory analysis defines in detail data describing the production process and the flows of materials and energy. In LCA, it is useful to distinguish between two types of data: foreground data and background data. Foreground data refer in large part to primary data collected at the orchard level, based upon information provided by enterprise technicians. The LCA software SimaPro v.9.0 accounted for the definition of background data and changes to the ecosphere, as a result of the production activities performed within the system boundary (Figure 1).

2.4.1. Foreground Data

Foreground data are reported in Table 2. Information regarding fertilization and application of agrochemicals were made available in orchard registers in which all cultivation measures along the

production season were noted. Water use was registered both as cultivation measure (fertigation) and as irrigation, monitored via dripper meters within drip irrigation systems at each sixth plant. Boxes with a tare weight of 2 kg (gross approximately 16 kg) and bins with a tare weight of 33 kg (gross approximately 230 kg) were used to collect fruit during the harvests. Both items are made of high-density polyethylene, with an estimated useful life of 15 years.

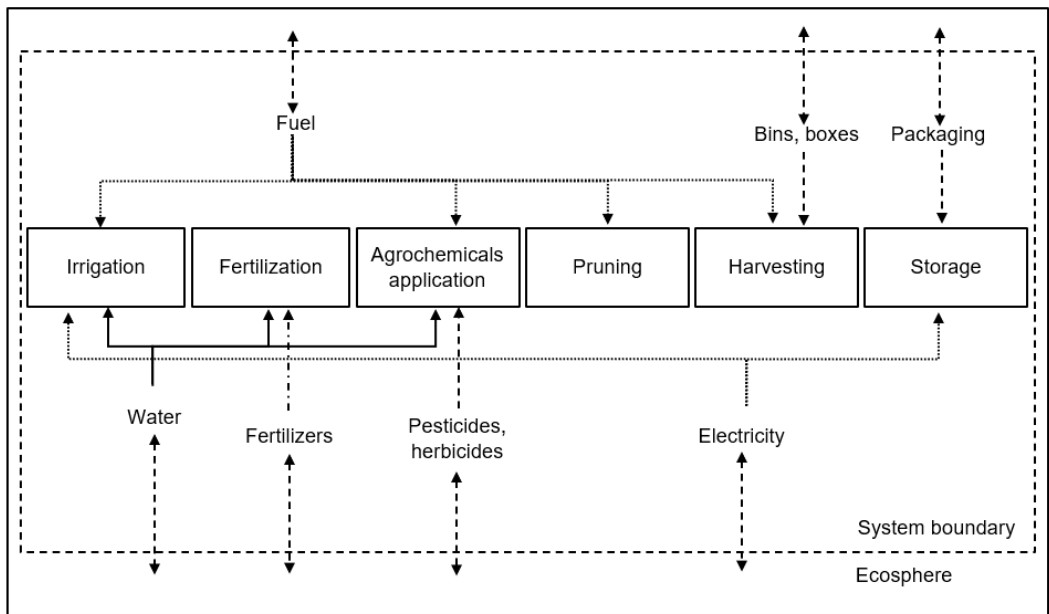

**Figure 1.** Main processes, input, and elementary flows included in the foreground system.

Enterprise technicians provided data regarding average fuel consumptions for agricultural machines, and the number of operations occurred in the orchards during the season. This information is summarized in Table 3. Pruning was carried out in a closed-loop cycle with biomass reintegrated, with positive contributions to soil organic matter improvement. Yearly pruning residues were quantified in the measure of 2 tons per hectare for every orchard and modeled as organic input to soil.

Electrical energy supplied from the national grid was used for irrigation and refrigeration; the main obstacle in reporting information consisted of the fact that there was no explicit data on energy consumption for the different production stages. For this reason, secondary data were estimated based on the following information provided by technicians: electrical energy for irrigation was estimated considering the water pump available of nominal power equal to 10.5 kW and average water mass flow equal to 640 liters per minute. Once harvested, fruits were stored in cold rooms. Since it was not possible to split energy data required for refrigeration by checking invoices, it was assumed that a weekly demand of 0.1 kWh per kilogram of fruit stored from a similar context [15], over the following average refrigeration time set out by farm technicians: 12.5 days for apricots, 4 for nectarines, 35 for plums, 90 for pears and kiwifruit, and 65 for apples. The refrigeration plants were ammonia ($NH_3$) based.

### 2.4.2. Foreground Data

Background data account for the production of input, energy, and waste management. They were taken from the international databases Ecoinvent v.3.6 and Agri-footprint v4.0, which include inventories of crop cultivation, data on transport, and agricultural input production for life cycle assessments. Background data collection was enabled within SimaPro v.9.1.

**Table 2.** Primary data collected.

| | Unit | Apricot | Nectarine | Plum | Pear | Apple | Kiwi |
|---|---|---|---|---|---|---|---|
| - Fertilizers | | | | | | | |
| Growth regulators | (kg ha$^{-1}$) | 7.01 | 3.65 | 11.35 | 28.80 | 2.26 | 8.20 |
| N-based | (kg ha$^{-1}$) | 331.19 | - | 137.40 | - | 148.97 | 1.50 |
| N-P-K-based | (kg ha$^{-1}$) | 534.49 | 0.55 | 119.48 | - | - | 201.75 |
| N-K-based | (kg ha$^{-1}$) | 14.83 | - | - | 17.10 | - | 7.02 |
| Mg-N-based | (kg ha$^{-1}$) | 19.65 | - | 107.53 | - | - | - |
| Microelements | (kg ha$^{-1}$) | 0.57 | - | 28.97 | 46.96 | 8.80 | 54.86 |
| Compost | (t ha$^{-1}$) | 20 | 20 | 20 | 20 | 20 | 20 |
| - Agrochemicals | | | | | | | |
| Insecticides | (kg ha$^{-1}$) | 19.35 | 14.54 | 65.6 | 60.44 | 13.27 | 21.54 |
| Fungicides | (kg ha$^{-1}$) | 16.01 | 23.12 | 59.6 | 57.87 | 23.23 | 6.00 |
| Herbicides | (kg ha$^{-1}$) | 5.03 | 2.47 | 2.17 | 11.80 | 4.40 | 2.66 |
| Other agrochemicals | (kg ha$^{-1}$) | - | 29.87 | - | - | 14.00 | - |
| - Water use | | | | | | | |
| Irrigation | (m$^3$ ha$^{-1}$) | 1633.33 | 2666.67 | 3266.67 | 3266.67 | 3984.00 | 4166.67 |
| Treatments | (m$^3$ ha$^{-1}$) | 9.17 | 18.54 | 63.60 | 78.39 | 168.16 | 97.00 |
| - Collection | | | | | | | |
| Boxes | (ha$^{-1}$) | 1413 | - | 3744 | 15 | - | |
| Bins | (ha$^{-1}$) | - | 120 | - | 99 | 303 | 89 |

**Table 3.** Use of agricultural machines in orchards.

| | Chopping | Fertilization | Pruning [1] | Agrochemical Application | Harvest |
|---|---|---|---|---|---|
| - Fuel consumption (L ha$^{-1}$) | | | | | |
| Tractor | 6 | 3 | 3 | 8 | 3 |
| + harvester wagon | - | - | 2.5 | - | 2.5 |
| + bin wagon | - | - | - | - | 2.5 |
| - No. orchard operations (year$^{-1}$) | | | | | |
| Apricot | 12 | 10 | 1 | 10 | 1 |
| Nectarine | 12 | 3 | 3 | 16 | 1 |
| Plum | 7 | 13 | 3 | 21 | 1 |
| Pear | 7 | 22 | 1 | 30 | 1 |
| Apple | 10 | 9 | 3 | 35 | 1 |
| Kiwi | 10 | 14 | 4 | 10 | 1 |

[1] 1/15 operation was considered to model plant eradication at the end of life (1/20 for kiwi).

## 3. Results

The elementary flows emerged within the inventory analysis were assigned in SimaPro v.9.1 to the following relevant impact categories: "Human health", "Ecosystem quality", "Climate change" and "Resources" according to the substances" ability to contribute to different environmental impacts. Particularly, the way environmental impact affects human health had been quantified with the metric Disability-Adjusted Life Years (DALY), expressed as the number of years lost due to illness, disability, or premature death [16]. "Ecosystem quality" encompasses multiple independent impact categories such as eutrophication, acidification, ecotoxicity, land use and water use, and it is measured in Potential Damage Fraction (PDF), defined as the fraction of species that have a high probability of not surviving in the affected area due to unfavorable living conditions [17]. "Climate change" describes the potential impact of different greenhouse gaseous emissions and it is measured in terms of carbon footprint (kg $CO_2$ eq.), while "Resources" models the primary energy needed (MJ) to made inputs available at

the technosphere. All these categories had been modeled in SimaPro v.9.1 with the life cycle impact methodology "IMPACT 2002+" [18].

### 3.1. Impact Assessment

The impact assessment was reported in terms of total environmental damage for humans according to the most known impact categories, as reported within Section 3.1.1. Then, it was allocated to each production activity considered (Section 3.1.2). The last Section 3.1.3, summarizes the impact for each production activity regarding the category "Climate Change".

### 3.1.1. Total Damage

The environmental impact of the four impact categories was normalized to depict a significant overview of the LCA results. Normalization is helpful to understand the relative weight of each category, making all units of measure compatible. As a result, total damage was split among impact categories as follows: 34.7% human health, 26.8% resources, 26.6% climate change, and 12.0% ecosystem quality (Figure 2). Human health was found to be the most impacting category for each type of fruit. The impact for resources was slightly higher than climate change for nectarines, pear, apple, and kiwi, but was lower, even in absolute terms, for apricots and plums. Ecosystem quality was the least impacting category. This result can be reasonably associated with the fact that fruit cultivation remains mostly a natural process characterized by relatively low industrialization. Overall, winter fruit is characterized by a higher impact compared to autumn and summer fruit, mainly because of the longer conservation time.

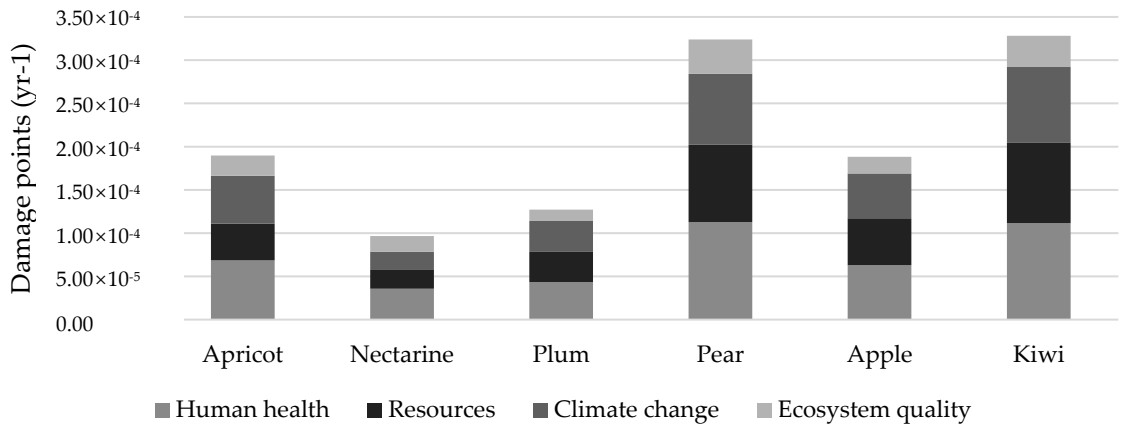

**Figure 2.** Normalized impact categories of fruit production.

The impact categories considered in this LCA were further divided into relevant subcategories (Figure 3). Other subcategories generally considered in environmental assessments, such as carcinogens, ozone layer depletion, respiratory organics, aquatic ecotoxicity, terrestrial acidification, land occupation and mineral extraction, resulted as not relevant.

### 3.1.2. Impact of the Production Processes

The processes considered in this LCA for the various fruits cultivated are characterized by different environmental impacts (Table 4). Planting activity is below 5%, with the highest value for nectarines. The impact of fertilizers is particularly high for apricots, while relevant for plums and kiwifruit; in this regard, the impact of organic fertilizer (compost) is negligible. Irrigation has a noticeable impact, with varying contribution between water and pumping energy, although energy impact is generally lower than water. Regarding agrochemicals, the overall impact is low, except for nectarines (6.41%) and apples (5.37%); insecticides and fungicides are likely to affect the environment jointly much more relevantly than herbicides, whose contribution is generally negligible. The impacts

associated with agronomic operations, due to consumption of fossil fuel, accounted for the highest contributions, between 1/5 and 2/5 of the total damage. Contribution from plastic use (harvesting boxed and packaging) was considered negligible, as roughly below 3%. It is evident that, for all fruit stored for a long time in cold rooms, refrigeration is the most impacting activity, with a relative impact up to half of the total damage for pears (53.55%).

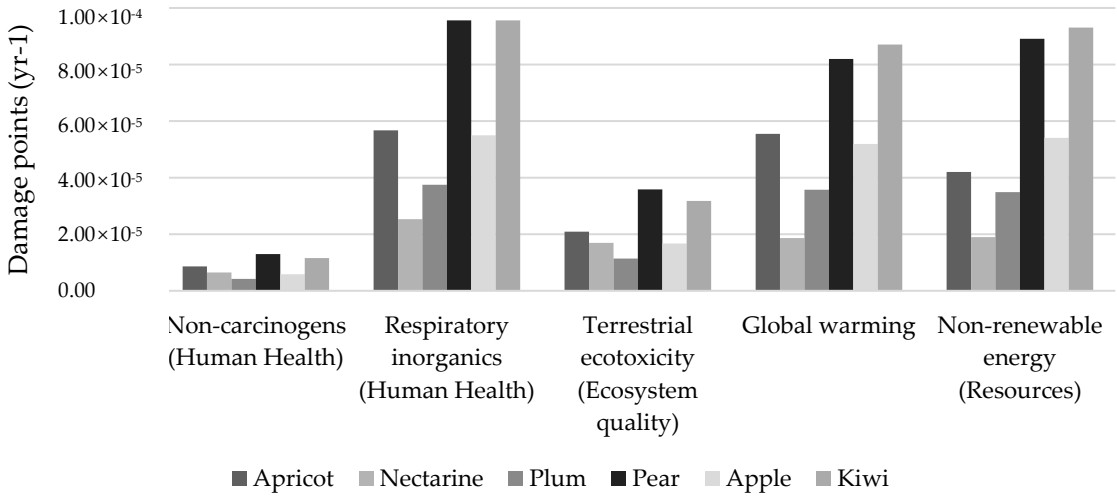

**Figure 3.** Normalized impact subcategories of fruit production.

Figure 4 shows the most impacting activities, thereby making it possible to identify the different contribution by each fruit. The highest impact for winter fruit (pears, apples, and kiwifruit) was due to refrigeration, while fertilizers for apricots and agronomic operations for nectarines, which is the least impacting fruit.

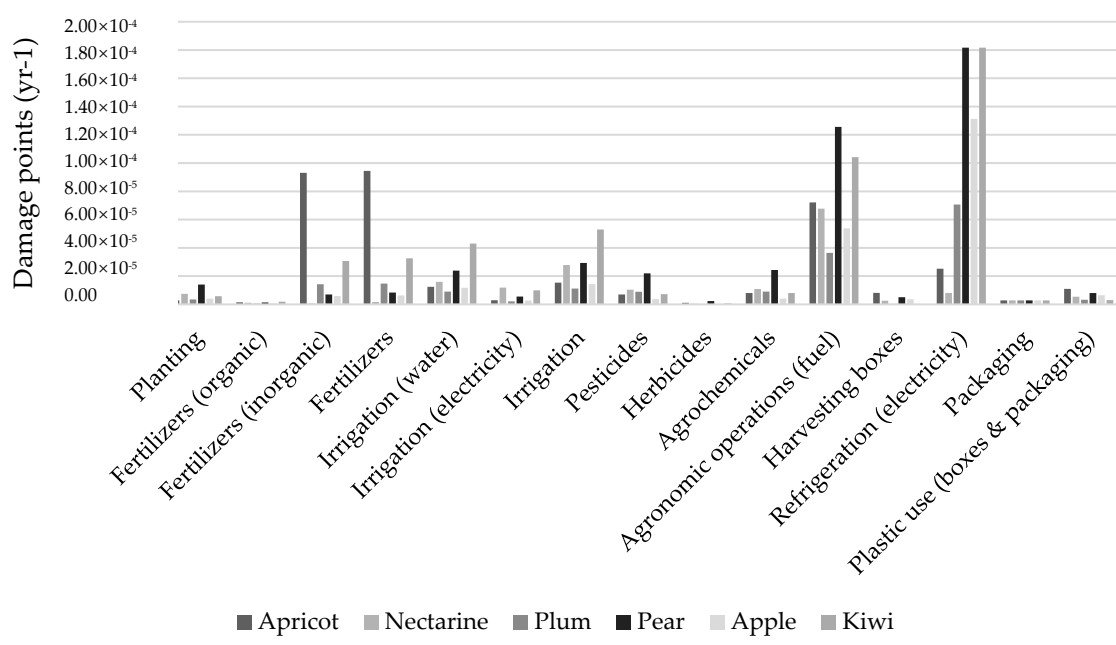

**Figure 4.** Most impacting processes for fruit production.

**Table 4.** Normalized impact of fruit production processes (yr$^{-1}$).

| | Apricot | Nectarine | Plum | Pear | Apple | Kiwi |
|---|---|---|---|---|---|---|
| - Planting | **0.82%** | **4.42%** | **1.87%** | **3.09%** | **1.66%** | **1.19%** |
| Fertilizers (organic) | 0.40% | 0.65% | 0.27% | 0.29% | 0.21% | 0.39% |
| Fertilizers (inorganic) | 26.83% | 0.36% | 7.73% | 1.55% | 2.38% | 6.38% |
| - Fertilizers | **27.23%** | **1.01%** | **8.01%** | **1.84%** | **2.59%** | **6.76%** |
| Irrigation (water) | 3.58% | 9.43% | 4.95% | 5.26% | 4.78% | 8.93% |
| Irrigation (electricity) | 0.84% | 6.97% | 1.15% | 1.21% | 1.08% | 2.06% |
| - Irrigation | **4.43%** | **16.40%** | **6.10%** | **6.48%** | **5.86%** | **10.99%** |
| Fungicides, insecticides, herbicides | 2.00% | 6.16% | 4.84% | 4.84% | 1.51% | 1.50% |
| | 0.31% | 0.25% | 0.09% | 0.53% | 0.14% | 0.16% |
| - Agrochemicals | **2.32%** | **6.41%** | **4.93%** | **5.37%** | **1.66%** | **1.66%** |
| - Agronomic operations (fuel) | **20.80%** | **39.97%** | **19.82%** | **27.71%** | **21.95%** | **21.64%** |
| - Harvesting boxes | **2.33%** | **1.53%** | **0.19%** | **1.13%** | **1.46%** | **0.04%** |
| - Refrigeration (electricity) | **7.27%** | **4.76%** | **38.48%** | **40.09%** | **53.55%** | **37.72%** |
| - Packaging | **0.82%** | **1.69%** | **1.56%** | **0.63%** | **1.17%** | **0.59%** |

### 3.1.3. Carbon Footprint

As highlighted within Section 2.3, the type of functional unit chosen (1 kg of fruit) is suitable for consumer value interpretations, potentially supporting consumer decisions in choosing products with lower environmental impact. Table 5 shows the carbon footprint associated with fruits considered.

**Table 5.** Carbon footprint (gCO$_{2eq.}$)—functional unit (FU): 1 kg of fruit.

| | Apricot | Nectarine | Plum | Pear | Apple | Kiwi |
|---|---|---|---|---|---|---|
| Planting | 3 | 7 | 3 | 13 | 4 | 6 |
| Fertilizers | 322 | 5 | 50 | 29 | 21 | 110 |
| Irrigation | 27 | 57 | 19 | 51 | 25 | 91 |
| Agrochemicals | 17 | 23 | 19 | 52 | 9 | 17 |
| Agronomic operations | 62 | 59 | 31 | 109 | 46 | 90 |
| Plastic use | 46 | 34 | 29 | 40 | 36 | 29 |
| Refrigeration | 72 | 23 | 201 | 516 | 373 | 516 |
| Remaining processes | 1 | 0 | 1 | 3 | 0 | 4 |
| Total | 550 | 208 | 354 | 811 | 514 | 862 |

## 4. Discussion

In this section, we discuss the results in terms of possible innovations regarding the production process (Section 4.1). Thus, we analyze the multiple ways in which value can be addressed under the lens of business model innovation (Section 4.2).

### 4.1. Incremental Innovation at the Process Level

#### 4.1.1. Mitigating the Impacts Associated with Refrigeration

The high impact accounting for fruit refrigeration emphasized the need to make the refrigeration process more efficient, less energy-intensive, and less dependent on non-renewable energy sources. Fruit depos in the case study were supplied by ammonia-based refrigeration plants, which constitute the majority of cooling units in developed countries. Ammonia is a common choice of refrigerant in large systems thanks to its thermo-dynamical properties that make it 3–10% more efficient than other refrigerants [19], and plants based on ammonia are some of the most diffused [20]. This natural refrigerant is toxic and flammable in large concentrations. However, it causes no ozone

depletion or global warming per se, unlike other refrigerants, such as chlorofluorocarbons (CFCs) and hydrofluorocarbons (HFCs).

The issue of refrigeration can be handled within an agricultural enterprise in different ways. One of the challenges in the refrigeration process is to use the already installed cooling capacity more optimally. In the last years, significant efforts in making refrigeration units more efficient are acknowledged. As an example, it was demonstrated that using closed coupled components in a compact refrigeration package, and electronic refrigerant injection control technology, facility ammonia charge could be reduced by more than 98%, with a related 7% reduction in energy and 3% reduction in water usage [19]. Compressors' optimal control was important to reduce refrigeration energy demand. Optimizing compressors' operation has led to minimizing power consumption while preserving the total refrigeration load requirement. A yearly monetary saving between €30,000 and €50,000 euros for eight refrigerating units, ranging in cooling capacity size from 180 kW to 795 kW, was estimated in [21]. Possible strategies for limiting the supply of refrigeration energy from fossil fuels is utilizing a combination of renewable energy sources, even limited to a portion of the refrigerating unit. Thermally driven absorption chillers within ammonia-based refrigeration units can be powered by low-grade heat or renewable energy resources. Such a practice can effectively represent a possibility to recover and re-use amounts of waste heat deriving from other industrial processes, improving in turn process efficiencies, and mitigating the associated environmental impacts. Significant industrial opportunities for the utilization of lower temperature heat from various industrial processes and energy sources are today available [22].

Another opportunity concerning sustainable energy concerns the integration of sustainable sources like solar photovoltaic (PV). PV systems are attractive because of their simplicity and low maintenance requirements [23]. They can be integrated directly into the unit they are providing power for, and can be mounted on roofs, without taking away agricultural land.

All the aforementioned strategies implicate a re-design of the refrigeration process, taking action on plants or plant portions, requiring business planning and investments over a low- to mid-term period. Other aspects involve a socio-cultural perspective, dealing with consumer preferences and choices. In the food sector, cold rooms are indispensable tools to ensure consumers with safe and genuine products: temperature is an essential requirement for the conservation of fruit (but also vegetables) and the preservation of nutritional properties. However, especially in case of extended supply chains, this mechanism seems to be heavily overused since producers are forced to anticipate the harvest, making fruit ripen in cold rooms. Furthermore, on the demand side, consumers are rather accustomed to the almost constant availability of the most widespread fruit varieties during the year. Mature consumers could perceive and evaluate seasonal fruit as more valuable of fruit ripened within cold rooms for a long time, being aware of a reduced environmental impact characterizing short food supply chain, in which production stage is more temporally and spatially close to consumption stage. A change in the demand side, even driven by virtuous food producers, would be desirable. In this regard, promising signs have been acknowledged in [24], in which it was observed that adopters of sustainability practices generally rate the benefits (measured as satisfaction level) higher than the investments in labor and financial resources.

### 4.1.2. Mitigating the Impacts Associated with Agricultural Operations

The high environmental impact of farming operations, from planting to harvesting poses a call for rationalizing farm operations. Enhanced connectivity and advanced data management can support substantial changes to increase efficiency. Since fuel consumption is proportional to in-field travelled distance, the use of technologies like auto-steering and section control can help in reducing unnecessary input applications, including fertilizers and agrochemicals, and thus fuel. Overlap reduction for auto-guidance systems may fall within 3–7%, with a proportional amount of agricultural inputs saved to the un-overlapped area [25]. In general, precision agriculture also encompasses a set of other novel technologies (e.g., remote sensing, variable-rate technologies) able to make agriculture more efficient,

thus mitigating the negative impacts of farming [26]. Information and Communication Technology applied to agriculture has enabled rapid, precise, and localized collection and processing of large pieces of data, generating suitable information for the daily operations of agricultural machinery. Auto-steering and section control on fertilizer spreaders appear to be viable solutions for many large-scale farms, and relevant economic benefits can arise from combining the use of the several Precision Agriculture technologies [27]. Process rationalization and digital switchover open numerous opportunities because they allow for process flexibility and manage complexity through data science. Digitization has become a fundamental path also for agricultural enterprises facing the growing complexity and uncertainty in their context. However, they represent changes in work culture rather than just technology, as they affect every worker, from enterprise managers to operators.

A new frontier in farm mobility is represented by agricultural machines powered with electric or hybrid motors. A wide number of agricultural machine manufactures are putting efforts on e-mobility development, and on the normation side, a new ISO technical standard is currently in progress and could be ready within the next three years. The adoption of electrical agricultural machinery will contribute to switching the environmental burden from the agricultural fuel supply chain to electricity generation and battery management. In this context, the possible advantages are twofold: on the one hand, energy supply can be delegated to renewable sources, and on the other hand, energy efficiency will be likely to increase, as the most diffused steel diesel engines show low overall energy efficiency (about 20%) [28].

### 4.1.3. Water Use: Room for Reduction?

Despite impacts due to irrigation were not associated with relevant environmental damage, the considerable amounts of irrigation water used is a relevant issue. The unit cost of water is low, and this may lead to the opportunistic behavior of irrigating much more than necessary. A possibility to come up with effective actions is to adopt comprehensive, integrated approaches for environmental management of water, improving water control capability and enhancing water supply predictability. Moreover, it would be appropriate to increase transparency and accountability in the context of agricultural enterprise, with water pricing based on measured deliveries [29].

### 4.1.4. Reducing Inorganic Fertilizers Use

The overall impact of fertilizers in orchards can vary from season to season. For the case study considered in this work, a particularly high impact was registered for the apricot orchard, while it is lower but still substantial for plums and kiwifruit. On the contrary, the impact of organic fertilizer (compost) was negligible for all orchards considered.

Farmers determine the amounts of nitrogen, phosphorous, and potassium fertilizers to be applied over the fields based on both their experience and the amounts applied the previous season, with maximum threshold levels based on worst environmental and climate situations delineated by regional agricultural policy administration. Since fertilizers are relatively cheap, often farmers prefer applying much more than required by fruit trees. Several negative effects on the environment are the consequence: to name but the most acknowledged, air pollution due to nitrogen emissions, water and soil degradation because of nitrogen leaching, eutrophication [26,30]. Precision agriculture can mitigate those negative effects through site-specific mapping and variable fertilizer application rates within plots. Satellite imaging or monitoring Unmanned Aerial Vehicles (UAVs) mounting N-sensors or multi-spectral cameras can sense cultivated fields and crops, and the deriving information can be processed to obtain fertilizer prescription maps, with the orchard divided into several areas characterized by a precise amount of fertilizer to be applied (kg/ha). Monitoring technologies are relatively cheap compared to traditional agricultural machines, and the information they make available can help reduce the impact of agronomic operations.

Other eco-sustainable solutions refer to conservation agriculture. The reduction of tillage and other farming operations can increase soil organic matter, while reducing costs, and can be applied to fruit growing, too.

*4.2. Business Model Innovation*

The use of the LCA framework has allowed highlighting several opportunities to make production processes cleaner and to reduce input use. Moreover, these opportunities can reflect in the way the agricultural enterprise addresses its own business canvas. All incremental innovations above described contributing to enhancing the value proposition, creation, and capture. Concerning value proposition, the proposed innovations for making production cleaner and more efficient can translate in a superior value that is offered to customers, with positive impacts for the society and the environment that can benefit from a reduced energy and input demand. In this way, the enterprise has the chance to strengthen consumer relationship, and address consumer decisions in choosing products characterized by a mitigated social and environmental burden.

Considering the definition provided in [31], meeting consumer expectations while reducing the environmental impact of the organization can be viewed as green marketing [32,33]. This approach fits particularly well with the context of fruit production, especially if the use of resources is reduced and if the enterprises adopt plans to reduce waste and pollution. Moreover, market communication of socio-economical aspects arising from LCA is viewed as part of sustainable marketing programs [11] and is associated with competitive advantage [34]. The benefits of green marketing can even go beyond the single company and embrace the whole value chain thanks to the reverse information flow and the importance of added-value services provided by intermediaries [35].

Value creation is concerned with making production activities and resource use more efficient. The call for agricultural inputs characterized by reduced impacts can, in turn, let suppliers and other partners of the supply chain be more aligned with the proposed innovative business model. Improvements in technical sustainability standards encourage quantification by the internal enterprise accounting for the impacts on ecosystems [36,37]. Apparently, this sounds like a virtuous cycle: sustainability assessments promote the internalization of ecosystems costs by embracing the growing focus on quantification of product sustainability, thereby influencing the sustainability standards of tomorrow. Value capture affects the enterprise cost structure and revenues. Economic benefits can result from monetary savings due to reduced agricultural inputs and by higher prices that customers would be willing to pay for more environmentally friendly products. These benefits must be counter-balanced by investments to improve technical efficiency. The economic impact of innovation uptake is a current research topic, with many promising findings highlighting the financial convenience, both on the cost reduction [27,38], and increased revenues [39] sides. Regarding revenues, the success of more environmental-friendly products depends on the adopted marketing policy able to satisfy and meet consumers' demand more effectively than competitors [39,40]. On the other hand, enterprises are required to deliver clear and pertinent product information through labeling and packaging; thus, increasing the perceived value and consumers' interest in buying products characterized by a mitigated environmental burden [11].

The above discussion shows that incremental innovations enhancing the environmental performance can benefit not just the enterprise that can make production more efficient and adopt green marketing strategies opening the possibility to increase in sales, but also the consumers themselves and the surrounding environment.

## 5. Conclusions

One of the urgent changes required to make agriculture more sustainable is to make production processes more efficient. The agricultural activities analyzed have highlighted several opportunities to reduce the environmental impact associated with fruit production in terms of efficiency improvement and input use reduction. Apparently, changes and improvements in production processes are

significantly relevant for those producers that utilize large amounts of natural inputs like land and freshwater, artificial inputs like inorganic fertilizers, and non-renewable energy sources.

Fresh food refrigeration is an urgent issue, especially for natural goods out of season. In addition, it would be important to invest in more efficient machinery and processes. On the other hand, enterprise communication strategies should encompass the lower environmental burden characterizing seasonal products, educating the public to more responsible consumption.

At the same time, there is room to reduce agricultural inputs such as water and fertilizers, as well as the energy required by in-field operations. However, we experienced in this case study that successful technology adoption also passes from the development of workers and employees, who are in the best position to identify the potential of technologies and direct them towards the optimization of the enterprise. People's skills, innovative capacity, and willingness to contribute are fundamental elements of a successful technology uptake process [40]. Enterprise managers, on the other side, have the delicate task of guiding technological change processes, triggering cultural transformation and management models.

Quantifying the environmental profile of production processes represent the first step for agricultural enterprises to reshape the business model and anticipate strategic business planning compared to competitors. A cleaner and more efficient production process can translate in a superior created value that is offered to customers, as well as a higher value captured by the enterprise. In most cases, costs related to the implementation of innovative practices can be economically affordable even for SMEs, and this can positively reflect into shared advantages with other food supply chain actors. Tools, such as LCA, can pave the way to bridge the gap between producers and consumers, building a strong relationship based on transparency and trust, and address consumer decisions in choosing products, with lower environmental impacts.

Future studies may focus on extending the analysis to other crops with similar farming and distribution characteristics, as well as on broader applications of environmental assessments, covering impacts on the whole supply chain, business innovation practices and consumer relationships. Another promising area of research is the analysis of other dimensions of sustainability, including economic impact categories (not just costing) and social aspects.

**Author Contributions:** Conceptualization, M.M., M.C., and M.T.; methodology, M.M.; validation, M.M.; formal analysis, M.M., M.C., and M.T.; investigation, M.M.; resources, M.C. and M.T.; data curation, X.X.; writing—original draft preparation, M.M. and M.C.; writing—review and editing, M.M. and M.C.; visualization, M.M.; supervision, M.C. and M.T.; project administration, M.T.; funding acquisition, M.T. All authors have read and agreed to the published version of the manuscript.

**Funding:** This research was funded by the Emilia-Romagna Region, Italy, within the RDP program 2014–2020, Measure 16.2.01.

**Acknowledgments:** This study is part of the project named "DIGIFRUIT—Toward digitalization of fruit production" participated by Granfrutta Zani SC, FAMOSA Srl, PeerNetwork Srl, and StudioMapp Srl. We thank Lorenzo Donati of Granfrutta Zani SC for the precious support by providing data, information, and for the time spent in explaining and showing fruit production processes.

**Conflicts of Interest:** The authors declare no conflict of interest. The funders had no role in the design of the study; in the collection, analyses, or interpretation of data; in the writing of the manuscript, or in the decision to publish the results.

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
