# Peer review of "Interpreting Environmental Impacts Resulting from Fruit Cultivation in a Business Innovation Perspective"

_sustainability, doi:10.3390/su12239793_

Round 1
Reviewer 1 Report
Dear Colleagues,
Thank you for a very nice formulated paper. I enjoyed reading it and made a few proposals on the attached version.
I would propose you to focus in the future to the crops with similar botanical characteristics and agronomy requirements. In addition to PA, you may also refer to Conservation Agriculture with three interlinked principles - no-till, mulch cover and crop diversification that provide ecosystem services in addition to improving soil structure and fertility.
Please also consider the use of solar panels as an alternative to electricity.
Good luck,
Hafiz
Author Response
Dear Colleagues,
Thank you for a very nice formulated paper. I enjoyed reading it and made a few proposals on the attached version.
I would propose you to focus in the future to the crops with similar botanical characteristics and agronomy requirements. In addition to PA, you may also refer to Conservation Agriculture with three interlinked principles - no-till, mulch cover and crop diversification that provide ecosystem services in addition to improving soil structure and fertility.
Please also consider the use of solar panels as an alternative to electricity.
Good luck,
Hafiz
Thank you very much for the appreciation and the useful comments. We have integrated the text with the insights regarding the use of conservation agriculture and solar PV in the discussion section and future direction for research within conclusions.
Reviewer 2 Report
Although the topic is timely and interesting, in its current form suffers from significant limitations that undermine its contribution to the literature and the practice. Below, I list some of the comments and suggestions I have for this manuscript.
- The paper does include a research question. It is difficult to understand the aim of the study, as also the structure is almost confusing. At the end of the Introduction section, you stated what the research emphasises, so you stated a sort of results, but the aim of the study is not clear.I suggest to consider. https://doi.org/10.1016/j.techfore.2020.120398
- I suggest rewriting the last paragraph of the introduction. The three contributions are not well articulated in the paragraph. I suggest for the improvement DOI: https://doi.org/10.1007/978-3-319-71062-4. 978-3-319-71062-4
- The review of the literature is not thorough, so the reader is not given an adequate background about the topic. Although the literature is not adequately written, it offers no new information and no new slant on the topic. Most of the content in the manuscript is well known by the readers of the journal. I suggest for the improvement. https://doi.org/10.1007/978-3-319-58538-3_177-1
- First and more important, now the discussion section is not well elaborated. It is not clear which results contribute to which theoretical stream and how you position your study’s results, in general, the existing studies. Although I can see some points where you did this, it needs to be more elaborated with a richer body of studies.
-- Implications for future research may also be included in the conclusion at the end. This research has article has created a lively discussion on so many issues that were hitherto unheard of and not addressed.
Author Response
Reviewer 2
Although the topic is timely and interesting, in its current form suffers from significant limitations that undermine its contribution to the literature and the practice. Below, I list some of the comments and suggestions I have for this manuscript.
- The paper does include a research question. It is difficult to understand the aim of the study, as also the structure is almost confusing. At the end of the Introduction section, you stated what the research emphasises, so you stated a sort of results, but the aim of the study is not clear. I suggest to consider. https://doi.org/10.1016/j.techfore.2020.120398
We clearly defined the research questions at the end of the introduction. Thank you for your help, we recognized that the aim was only partially explained, as most of the emphasis was put on the LCA.
- I suggest rewriting the last paragraph of the introduction. The three contributions are not well articulated in the paragraph. I suggest for the improvement DOI: https://doi.org/10.1007/978-3-319-71062-4. 978-3-319-71062-4
Maybe there has been a mistake in the attached DOI, we have referred to the following one: https://doi.org/10.1007/978-3-319-71062-4. Anyhow, it is not clear for us which ‘three contributions’ the reviewer refers to. In the paragraph there are no cited references. We may think to the three dimensions of sustainability recalled also in the DOI above reported. In this case the paper mostly focused the environmental dimension of sustainability, as stated also in the manuscript title.
- The review of the literature is not thorough, so the reader is not given an adequate background about the topic. Although the literature is not adequately written, it offers no new information and no new slant on the topic. Most of the content in the manuscript is well known by the readers of the journal. I suggest for the improvement. https://doi.org/10.1007/978-3-319-58538-3_177-1
We have improved the review of literature, taking also account of your suggestion.
- First and more important, now the discussion section is not well elaborated. It is not clear which results contribute to which theoretical stream and how you position your study’s results, in general, the existing studies. Although I can see some points where you did this, it needs to be more elaborated with a richer body of studies.
In this work we have discussed all the inefficiencies and production bottlenecks highlighted by the LCA, exploring possible business model innovation stemming from an environmental assessment related to fruit production. Our aim was to provide practical evidence with a case study. We started from a life-cycle assessment to highlight production inefficiencies, and then we have extended the scope of environmental management to a business strategy view, in line with a sustainable development path for the agri-business. So, we have slightly adjusted section 4.1 (also according to Reviewer 1 feedback) and more deeply elaborated section 4.2 concerning business strategy. We have added discussion on green marketing approach, labelling and packaging considerations.
- Implications for future research may also be included in the conclusion at the end. This research has article has created a lively discussion on so many issues that were hitherto unheard of and not addressed.
Thank you for your precious revision. We have integrated some implications for future research as suggested by Reviewer 1. We believe that deepening the many issues treated in this work at once will fall beyond the scope of this work, which is devoted to the description of a specific case study regarding fruit production. We can deepen a particular issue if required.
Round 2
Reviewer 2 Report
-
-The idea of the paper is well in time, and it has the potential to contribute. However, I have some comments that will help to improve the quality of the manuscript..
The authors have addressed the majority of my comments at the previous round however, there are still some slight corrections that I recommend making before accepting the paper for journal publication. That does not mean that this is not a valuable and concise review of the topic, although it might raise a question about whether this journal is where it belongs.
- The Introduction and Background sections provide useful information for the readers. Nevertheless, some information presented is not accurate. The author also needs to justify the importance of understanding the vision of the topic.It would be nice to see a stronger connection between your findings and the theme of the journal. How does this understanding help organizations and the industry to make better decisions? The managerial and theoretical implication is missing at the moment. Thus the reviewer cannot conclude if the manuscript has contributed fully to the existing body of knowledge.
https://doi.org/10.1016/j.techfore.2020.120398
Role of Design Thinking and Biomimicry in Leveraging Sustainable Innovation
https://doi.org/10.1007/978-3-319-71059-4_86-1
Steering for Sustainable Development Goals: A Typology of Sustainable Innovation
DOI:
https://doi.org/10.1007/978-3-319-71059-4_64-1
Exploring the effect of buyer engagement on green product innovation: Empirical evidence from manufacturers,
DOI: https://doi.org/10.1002/bse.2631
- The overall quality of the paper is good, but according to my opinion, you must restrain this article to only one research question.
Minor General Comments
- The manuscript is potentially original contributive but needs a minor revision.
- Implications for future research may also be included in the conclusion at the end. This research has article has created a lively discussion on so many issues that were hitherto unheard of and not addressed.
- How the results of this study can be generalized to other companies.
-Also explain briefly what the future research opportunities are.
Author Response
Please find attached our letter
